# Characterization of microRNA Levels in Synovial Fluid from Knee Osteoarthritis and Anterior Cruciate Ligament Tears

**DOI:** 10.3390/biomedicines10112909

**Published:** 2022-11-12

**Authors:** Laura Rizzi, Marco Turati, Elena Bresciani, Filippo Maria Anghilieri, Ramona Meanti, Laura Molteni, Massimiliano Piatti, Nicolò Zanchi, Silvia Coco, Francesco Buonanotte, Luca Rigamonti, Giovanni Zatti, Vittorio Locatelli, Robert J. Omeljaniuk, Marco Bigoni, Antonio Torsello

**Affiliations:** 1School of Medicine and Surgery, University of Milano-Bicocca, 20900 Monza, Italy; 2Department of Paediatric Orthopedic Surgery, Hospital Couple Enfants, Grenoble Alpes University, 38400 Grenoble, France; 3Residency Program in Orthopaedics and Traumatology, University of Milan, 20122 Milano, Italy; 4Department of Orthopedic Surgery, Policlinico San Pietro, 24036 Ponte San Pietro, Italy; 5Transalpine Center of Pediatric Sports Medicine and Surgery, Hospital Couple Enfant, 38700 Grenoble, France; 6Orthopedic Department, San Gerardo Hospital, 20900 Monza, Italy; 7Department of Biology, Lakehead University, Thunder Bay, ON P7B 5E1, Canada

**Keywords:** osteoarthritis, anterior cruciate ligament, microRNAs, synovial fluid

## Abstract

This study investigated modifications of microRNA expression profiles in knee synovial fluid of patients with osteoarthritis (OA) and rupture of the anterior cruciate ligament (ACL). Twelve microRNAs (26a-5p, 27a-3p, let7a-5p, 140-5p, 146-5p, 155-5p, 16-5p,186-5p, 199a-3p, 210-3p, 205-5p, and 30b-5p) were measured by real-time quantitative polymerase chain reaction (RT-qPCR) in synovial fluids obtained from 30 patients with ACL tear and 18 patients with knee OA. These 12 miRNAs were chosen on the basis of their involvement in pathological processes of bone and cartilage. Our results show that miR-26a-5p, miR-186-5p, and miR-30b-5p were expressed in the majority of OA and ACL tear samples, whereas miR-199a-3p, miR-210-3p, and miR-205-5p were detectable only in a few samples. Interestingly, miR-140-5p was expressed in only one sample of thirty in the ACL tear group. miR-140-5p has been proposed to modulate two genes (BGN and COL5A1100) that are involved in ligamentous homeostasis; their altered expression could be linked with ACL rupture susceptibility. The expression of miR-30b-5p was higher in OA and chronic ACL groups compared to acute ACL samples. We provide evidence that specific miRNAs could be detected not only in synovial fluid of patients with OA, but also in post-traumatic ACL tears.

## 1. Introduction

Knee osteoarthritis is a chronic-degenerative joint disease with an estimated prevalence in the general population of 7.9% (CI 95%: 7.6–8.2%) [1] and an exponential increase after the age of 50. OA has multiple etiologies, including trauma as well as interrelated biochemical, biomechanical, and genetic factors [2].

In particular, anterior cruciate ligament (ACL) tears are believed to lead to post-traumatic osteoarthritis (PTOA), with a reported incidence as high as 87% [3], accompanied by a 30-year aging effect. [4]. These long-term effects are particularly concerning in light of the estimated 200,000 cases of ACL injury per year in the United States alone [5,6].

PTOA represents a significant and increasing cause of functional disability in youth as primary injuries are more likely to occur in younger individuals, participating in sports. PTOA is a progressive disease, and there is a compelling need to improve diagnostic techniques for its early detection. Presently, there are no approved clinical protocols to prevent traumatic injury from progressing to PTOA. Therefore, a better understanding of specific inflammatory biomarkers in the knee articular environment is vital. 

The exact mechanisms responsible for cartilage disruption and subsequent progression to knee OA following ACL injury are manifest but far from being fully understood. ACL injury and deficiency often results in knee instability that, by disrupting articular biomechanics, causes premature wear and tear [7]. Nevertheless, mechanical factors alone are not sufficient to explain the relationship between ACL rupture and PTOA. In fact, the incidence of PTOA remains high even after ACL reconstruction, an intervention that significantly improves knee stability.

In recent years, research has identified the triggering of specific inflammatory patterns as another possible driving factor [8,9]. In fact, any injury to the knee joint, such as ACL tears, impairs the synovial levels of substances implicated in joint homeostasis and degeneration [10,11,12].

In addition to the upregulation of many inflammatory cytokines, current research has revealed prominent roles of microRNAs, lncRNAs, and exosomes in modulating disease progression. The role of microRNAs both in maintaining tissue homeostasis and in determining the pathological changes occurring during joint diseases has been hypothesized [5,13]. With regard to the pathogenesis of OA, microRNAs are able to regulate chondrocyte apoptosis and proliferation, extracellular matrix metabolism, and the inflammatory response. In a recent study, a group of 16 microRNAs appeared to be involved in inhibiting the OA process, whereas 14 different microRNAs appeared to promote OA. Previous research focused on microRNA expressed at the tissue level [13] as well as in circulating cells [14]; by comparison, there is scant data on microRNA expression in synovial fluid (SF).

SF is relatively easy to collect and rich in biological information, and has been used in studies to explore OA pathogenesis. SF is abundant in active soluble molecules that can stimulate cell responses directly, as in the case of cytokines, or indirectly through extracellular vesicles containing microRNAs. The identification of novel biomarkers of OA disease risk and/or progression could allow the discovery and validation of cellular and molecular targets for novel therapies to prevent and treat PTOA [15].

The primary goal of the present study was to evaluate differential expression of selected miRNAs in SF of patients with knee osteoarthritis and patients with an isolated ACL rupture. We evaluated 12 microRNAs that have been considered relevant to the pathological process in bone, cartilage, and ligaments.

## 2. Materials and Methods

### 2.1. Subjects

Forty-eight subjects (28 males and 20 females) were enrolled in the study and resolved into two groups based on their diagnosis. Group A included patients ranging from 12 to 50 years of age, with simultaneous joint effusion, isolated traumatic anterior cruciate ligament rupture, arthroscopically confirmed. Exclusion criteria included: (i) Previous trauma of the limb, (ii) articular cartilage injury greater than Outerbridge Grade I, (iii) medial collateral ligament lesions greater than Grade I, (iv) previous injury to the medial or lateral meniscus and posterior cruciate ligament, (v) infective or inflammatory diseases of the knee, (vi) previous intra-articular injection of steroid or hyaluronic acid, and (vii) time from injury greater than 90 days. Group B included patients ranging from 18 to 85 years of age, with radiologically confirmed knee osteoarthritis with an end-stage OA (Grade ≥3 according to the Kellgren-Lawrence classification) [16]. Exclusion criteria included: (i) Simultaneous articular inflammatory disease, (ii) chondrocalcinosis, (iii) simultaneous tumors, (iv) intra-articular injection of steroid or hyaluronic acid in the previous 6 months, (v) infective disease involving the knee or surrounding tissues, and (vi) dermatological disease of the withdrawal site. Group A was composed of 30 patients, including 19 males and 11 females; Group B was composed of 18 patients, including 9 males and 9 females. 

Group A contained 13 patients with hydrarthrosis, 2 with mild hemarthrosis, and 15 with hemarthrosis; by comparison, Group B contained 14 patients with hydrarthrosis, 1 with mild hemarthrosis, and 3 with hemarthrosis. Samples of patients in Group A were collected at various times following onset of the disease: Six were collected in the acute phase (defined as 0–2 days after injury), three in the early subacute phase (3–15 days after injury), and twenty-one in the late subacute phase (15–90 days after injury). Patients in Group A were also resolved into 2 subgroups according to age and skeletal maturity: (i) Adolescents and (ii) adults. The adolescent subgroup consisted of patients between 10 and 17 years with a traumatic ACL tear (16), whereas the adult subgroup consisted of patients ≥18 years of age with a confirmed isolated ACL tear [17]. Samples of Group B were all defined as chronic, since they were all collected >90 days after disease onset (Table 1).

All patients signed a written informed consent to the retention of biological material that would have otherwise been discarded. The experimental protocol was approved by the local Ethical Committee and conforms to the principles outlined in the WMA Declaration of Helsinki.

### 2.2. Samples

Synovial fluid was aseptically drawn from the knee without any lavage, obtaining a volume of at least 1 mL. The procedure was carried out at different times: Some samples were collected at the time of the first medical examination in an Emergency Room, some in a medical office, and the most of them at the beginning of arthroscopic surgery. For patients drawn in ER and in a medical office, local anesthesia, using lidocaine or difluoroethane spray, was applied prior to sample collection. Patients sampled before surgery were already under local-regional anesthesia. Local trichotomy with a single-use razor and asepsis with iodopovidone were performed in all cases; 20-gauge by 1-inch-long needles were used for all sampling. Synovial samples were collected into tubes containing EDTA and immediately centrifuged at 3000× *g* to remove cellular debris. Then, the supernatant was stored at −80 °C until assayed [18,19,20,21]. 

### 2.3. The microRNA Extraction, cDNA Synthesis of microRNA, and Real-Time PCR

The microRNAs were purified using a flexible, organic-solvent-free isolation of small RNA kit (High Pure miRNA Isolation Kit, Roche, no. 05080576001) in accordance with the two-column protocol. In brief, in the first step, large molecules are bound to the silica matrix of the first column and thus removed from the sample. The flow-through is combined with Binding Enhancer and the second spin column was used to isolate the enriched miRNA fraction. For each patient, we used 150 μL of SF. For cDNA synthesis, we used TaqMan Advanced miRNA Assay (Thermo Fisher Scientific, Waltham, MA, USA, no. A25576) and Taqman Advanced miRNA cDNA synthesis Kit (Thermo Fisher Scientific, no. A28007) in accordance with the protocol. The specific quantification of mature microRNAs was performed using qPCR Applied Biosystems TaqMan Advanced miRNA Assays. The TaqMan Advanced miRNA cDNA Synthesis Kit uses 3′ poly-A tailing and 5′ ligation of an adaptor sequence to extend each end to the mature miRNAs in the sample, prior to reverse transcription. Universal RT primers recognized the universal sequences present on both the 5′ and 3′ extended ends of the mature miRNAs (Table 2). We chose to study these 12 miRNAs since it has been reported that they are involved in degenerative processes relevant for this study. In fact, bone metabolism is a multifactorial process that involves several cell populations, such as bone marrow mesenchymal stem cells, osteoblasts, osteoclasts, as well as osteocytes. In these cells, the activation of key signaling pathways is present, which can be modulated by modifications of the 12 miRNA levels selected in this research. All mature miRNAs in the sample were reverse transcribed to cDNA. For each different miRNAs, all the samples were analyzed using the ABI PRISM 7900 HT Sequence Detection System. The Ct (cycle threshold) is defined as the number of cycles required for the fluorescent signal to exceed the background level (i.e., the lower the Ct value, the more PCR product is present). Ct levels are inversely proportional to the amount of target nucleic acid in the sample (i.e., the lower the Ct level, the greater the amount of target nucleic acid in the sample). The delta-delta CT method (ΔΔCT) was used to calculate fold expression levels for each target gene as indicated in the protocol.

### 2.4. Statistical Analysis

Statistical analysis was performed using GraphPad Prism 7 (Dotmatics). Normality of the data was investigated using the Shapiro–Wilk test. Data were found to be non-normally distributed, and therefore non-parametric statistic tests have been performed. The influence of a variable of interest on microRNA levels was investigated through the Mann–Whitney test when comparing two groups, or the Kruskal–Wallis non-parametric test for three or more groups. When statistical significance was found, the Dunn test was performed. Correlations among biochemical markers were assessed for significance using the non-parametric Spearman rank correlation coefficient test.

For all statistical tests, a value of *p* < 0.05 was considered to be significant.

## 3. Results

### 3.1. Expression of microRNA

In this study, we measured the expression of all 12 microRNAs in synovial fluids obtained from the 48 subjects with RT-qPCR, obtaining 576 Ct values (Ct). Ct values ranged from 0 (indeterminate) to 40 (Table 3).

Levels of microRNAs were measured in synovial fluid by RT-qPCR. Ct values = 0 indicate the absence of the microRNA in the sample, whereas Ct > 0 suggest that those microRNAs were expressed in synovial samples.

First, the levels of microRNA expression in the synovial fluid were generally very low, and some were undetectable in many samples. Only three microRNAs, i.e., miRNA 26a-5p, miRNA 186-5p, and miRNA 30b-5p, were detectable in more than 50% of the samples. In particular, miRNA 26a-5p was detectable in 15 of 30 subjects in Group A; miRNA 186-5p in 17 of 30 subjects in Group A (knee OA), and in 14 of 16 subjects in Group B (ACL tear); miRNA 30b-5p in 12 of 18 subjects in Group B (Figure 1). Interestingly, miRNA 210-3p was detectable in 22% of OA samples but only 3% (1 subject) of ACL group. This is consistent with the potential role of miRNA 210-3p as a marker to the early diagnosis of OA in suspected individuals [22]. 

### 3.2. Expression of microRNA in Knee OA Group and ACL Tears Group 

After assessing the normal distribution of microRNA levels using the Shapiro–Wilk test, the two groups were further compared using the Mann–Whitney non-parametric test.

Levels of both miRNA 30b-5p and miRNA 140-5p were significantly higher (*p* < 0.05) in subjects with knee osteoarthritis. No statistical difference was calculated in the expression of the other miRNAs between Group A and Group B (Figure 2).

For many miRNAs, the characteristic “boxes” are not visible in the boxplot diagram since the majority of the values are zero and are therefore “squashed” on that particular value. Specifically, miR*-*let7a-5p was not present in 90% of samples of the ACL group and 89.9% of samples of the OA group. This result is surprising, since miR*-*let7a-5p has been proposed to regulate skeletal development by orchestrating the proliferation and differentiation of chondrocytes [23]. 

The miRNA 146 is thought to be among the mediators of inflammation; however, miR-146-5p was not detectable in 86.7% and 88.9% of samples from ACL and OA groups, respectively. Similarly, miR-205-5p was undetectable in 90% and 89.9% of samples. In accordance with the literature, we selected some of the samples for further investigation. We tested miRNA 26a-5p, miRNA 186-5p, and miRNA 30b-5p, which were expressed in more than 50% of the samples of each group (Table 4).

In the ACL group, 10 microRNAs were detectable in both the adolescent and adult groups and their levels were similar in both groups. 

### 3.3. Gender Influence on microRNA Expression

The influence of gender on the expression of miRNAs 30b-5p, 186-5p, and 26a-5p was first assessed considering the synovial fluid obtained from male subjects vs. females, regardless of their pathology. The results demonstrate that gender did not significantly affect expression of the selected miRNAs in knee synovial fluid (Figure 3). 

Moreover, a further stratification was performed to compare male vs. male and female vs. female subjects both in Group A and Group B. The results showed that miRNA 186-5p was expressed at higher levels in females with knee OA (*p* < 0.05) compared to females with ACL tears. This result is not surprising since circulating miRNA 186-5p has been proposed to be associated with knee OA in women. In contrast, miRNA 30b-5p was expressed at higher levels in males with knee OA (*p* < 0.05) compared to males with ACL tears (Figure 4).

### 3.4. Duration of the Disease Influence on microRNAs Expression

The expression of miRNA 30b-5p was influenced by the time from the onset of the disease. In particular, miRNA 30b-5p was expressed at higher levels in late subacute ACL injuries (*p* < 0.005) and knee OA (*p* < 0.05) compared to acute/early ACL injury. No differences were found for the other miRNAs tested (Figure 5).

### 3.5. Correlation between microRNAs

The miRNAs which presented positive Ct values, close to 50%, were further assessed for possible correlations in expression. These correlations, in light of the non-normal distribution of the data, were evaluated through Spearman’s correlation index for ranks. In OA patients, the expression of miRNA 186-5p was positively correlated with miRNA 30b-5p (*p* < 0.05). No other correlation was found among the expression of the other miRNAs (Figure 6).

## 4. Discussion

The 12 miRNAs we selected to investigate in this study were previously implicated in specific cellular signaling patterns. Some of these miRNAs modulate inflammatory responses (miRNA 26a-5p) [24] and cell proliferation (miRNA 26a-5p, miRNA 140-5p) [25,26], while others are implicated in chondrocyte homeostasis and differentiation (miRNA 30b-5p) [27]. Although some studies suggested that miRNA 210 [22,28] and miRNA 186-5p [23] could participate in the development of OA, the specific role of these two miRNAs remains largely unknown. According to our results, miRNA 26a-5p was the most expressed (45.6%) in synovial samples. It is known that downregulation of miRNA 26a-5p leads to a missed suppression of gene FUT4 with the consequent activation of the NF-kB pathway, and increased iNOS production. Furthermore, miRNA 26a-5p plays an important role in the regulation of some metalloproteases (ADAMTS-5 and MMP13), which are involved in ECM degradation and in aggrecan and collagen type II production [24,29,30,31]. Although our study lacked a control group with healthy patients (for ethical reasons), our results suggest the activation of common inflammatory pathways in ACL lesions and OA. In samples from the ACL tear group, miRNA 140-5p was significantly overexpressed compared to those in the OA group (3.3% vs. 27.8%, *p* < 0.05). This miRNA is a positive modulator of cellular proliferation of the growth plate [18]. However, the real role of miRNA 140-5p in intraarticular pediatric knee lesions, such as ACL, meniscal, and chondral tears has not yet been defined [32]. 

Studies on animal models highlighted the capability of miRNA 140-5p to stimulate stem cell differentiation into chondral cells [33]. Furthermore, miRNA 140-5p modulates inflammatory processes, inactivating the TLR4/Myd88/NF-kB pathway. This miRNA was not expressed in 72.5% of OA samples, consistent with the reported data. This phenomenon may be related to its modulation of two genes (BGN and COL5A1100), which are involved in ligamentous homeostasis [34]. The altered expression of BGN and COL5A1100 is probably linked to susceptibility for ACL rupture. The suppression of miRNA 140-5p expression after an ACL injury could induce an increase in fibrillary collagen production as a joint response to promote tissues repair. Willard et al. hypothesized that miRNAs may play a key role in the regulation of cell signaling in response to the mechanical loading and healing process in ECL tissue; however, more pre-clinical studies are needed to better explain these mechanisms [34,35,36,37]. 

Conversely, miRNA 30b-5p was overexpressed in the OA group compared with the ACL tear group (66.7% vs. 43.3%, *p* < 0.05). This miRNA is involved in the regulation of ERG (a subfamily of ETS transcription factors), and modulates different processes, such as embryonic development, cell proliferation and differentiation, angiogenesis, inflammation, and apoptosis. Furthermore, several studies report the involvement of miRNA 30b-5p in the downregulation of COL2a and aggrecan [27]. Increased miRNA 30b-5p levels in ACL injury could be a consequence of the inflammatory response. On the other hand, overexpression of miRNA 30b-5p in OA may play a role in cartilage degradation. Interestingly, the expression of miRNA 30b-5p in the ACL tear group varied with the time elapsed between injury and sampling. miRNA 30b-5p was significantly less expressed (*p* < 0.05), nearly absent, in samples collected in the acute phase (0–15 days after injury, ACL^A/ESA^ group) compared to the samples collected in the subacute phase (16–90 days after injury, ACL^LSA^ group). We found the same difference between the ACL^A/ESA^ group and OA group (*p* < 0.005). These findings suggest that the duration of time elapsed since the traumatic event is crucial to the expression of this miRNA. This could be explained by joint instability and mechanical stress of the knee during recovery after injury. Interestingly, no differences in miRNA 30b-5p levels were found between the ACL^LSA^ and OA group; this finding is consistent with the relationship between mechanical and biological factors resulting in inflammatory environments after an injury and OA development [15,27]. With regard to the influence of gender, we found a difference between the ACL and OA groups (*p* < 0.05) in males, which was not detectable in females. This dichotomy could be due, in part, to a different mechanism of injury between males and females, as well as a gender dependent difference in hormonal influence on the expression of miRNAs, as previously reported [38].

Gender influences were found for miRNA 186-5p levels, as well. In fact, this miRNA was significantly overexpressed in females with OA compared to females with an ACL injury (*p* < 0.05). The role of miRNA 186-5p remains unclear, although some studies have described it as a potential regulator of pathways, such as signaling by PDGF, developmental biology, membrane trafficking, and collagen formation. Although there are no studies on correlations between the presence of this miRNA in the joint and female gender, our results are consistent with a recent publication, in which serum levels of miRNA 186-5p were significantly higher in post-menopausal women with a diagnosis of OA [39]. Furthermore, a moderate positive correlation was found between miRNA 30b-5p and miRNA 186-5p levels (*p* < 0.05); however, the role of miRNA186-5p in the joint is still unclear.

Some recent studies suggest differences in the biochemical environment in skeletally immature patients in comparison to adults after an ACL tear [9,40]. Resolvin E1 and IL-10 levels are higher in skeletally immature ACL tears compared to adults; however, the real role of the microenvironment surrounding the ACL remains undefined. Interestingly, we observed that 10 miRNAs were detectable in both adult and adolescent groups with ACL tears without a meniscal tear. More pre-clinical studies are needed to better understand the intraarticular biochemical process in the healing potential of ACL tears in skeletally immature patients [41,42,43,44,45]. 

In conclusion, our results show a common expression pattern of the 12 miRNAs analyzed (26a-5p, 27a-3p, let7a-5p, 140-5p, 146-5p, 155-5p, 16-5p, 186-5p, 199a-3p, 210-3p, 205-5p, 30b-5p) between subjects with ACL injury and knee OA. Different bimolecular factors are involved in the development of post-traumatic osteoarthritis and probably some of them are present since the moment of injury. This study highlights some interesting aspects in the expression of miRNAs in ACL injury and in knee OA in relation to different factors, such as sex, gender, and time since the onset of the disease. 

Our results suggest that further studies carried out on a larger population are still needed. Finally, to better qualify the significance of our findings, additional knowledge of the complex pathways regulated by the miRNAs and the capability of each miRNA to impact on the expression of others is required.

We acknowledge the limitations of the present study. One limitation is the small number of miRNAs tested; although we chose them on the basis of their role in OA development, there are several other miRNAs that could play an important role in this disease. Another important limitation is the small number of subjects enrolled in the study, which could have influenced the statistical power of our analysis. However, patients were rigorously selected: The group with ACL tears presented only patients with a traumatic ACL tear confirmed by MRI and arthroscopically, without any chondral injury greater than Grade I according to the Outerbridge classification, and without any meniscal or ligament tear. Additionally, as there was no healthy control group in our study, we could not define a “baseline” in the expression of miRNAs considered. Performing an arthrocentesis in a healthy, non-swollen knee, without causing any damage is technically demanding. This very invasive technique is acceptable only when clinically justified and is not an ethically acceptable technique for a healthy knee. 

## Figures and Tables

**Figure 1 biomedicines-10-02909-f001:**
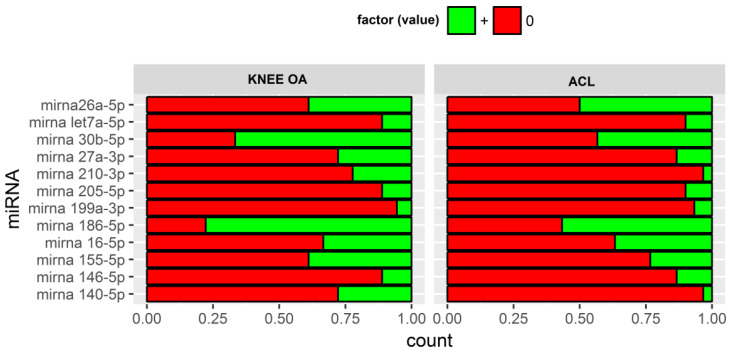
Expression of microRNAs. Percentage representation of null and positive samples in the two groups.

**Figure 2 biomedicines-10-02909-f002:**
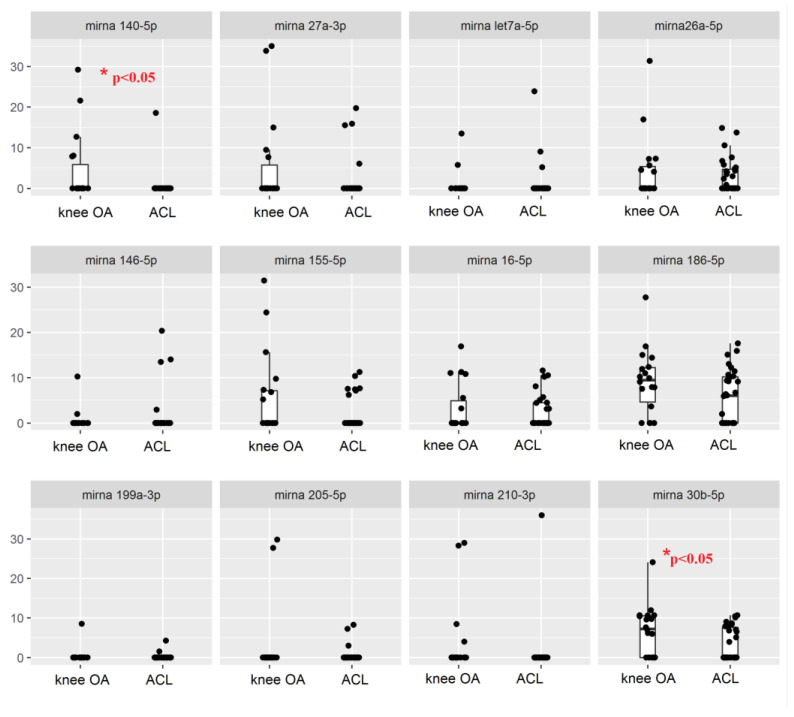
Boxplot comparing microRNA Ct levels in Gonarthrosis and ACL.

**Figure 3 biomedicines-10-02909-f003:**
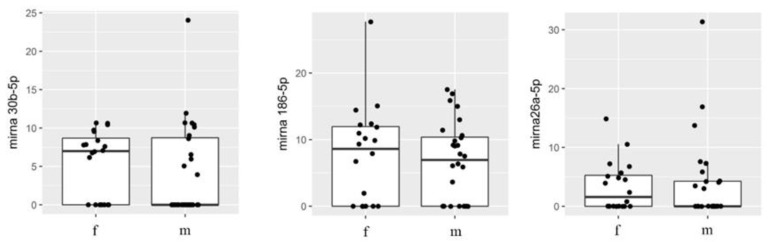
Boxplot comparing microRNA in females and males.

**Figure 4 biomedicines-10-02909-f004:**
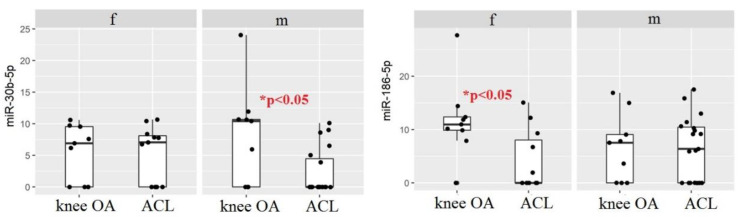
Boxplot comparing microRNA in females and males for the two groups of diseases.

**Figure 5 biomedicines-10-02909-f005:**
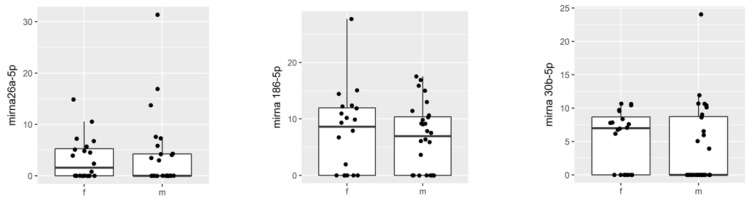
Boxplot of microRNA in ACL ^A/ESA^, ACL ^LSA^, and knee OA.

**Figure 6 biomedicines-10-02909-f006:**
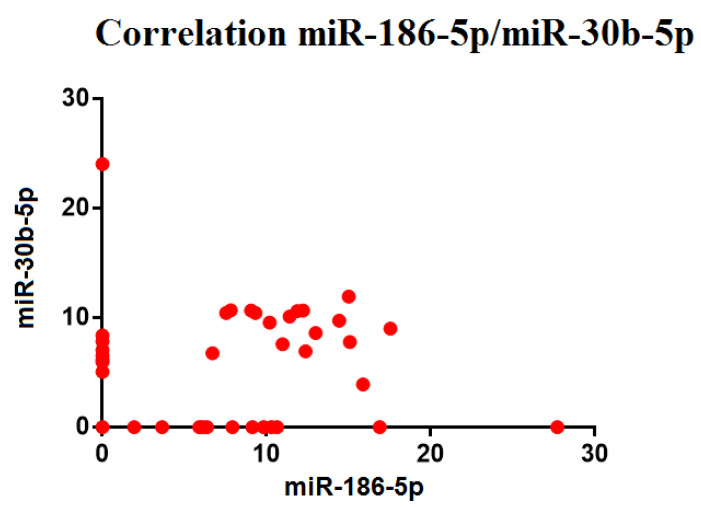
Correlation between miRNAs with statistical significance.

**Table 1 biomedicines-10-02909-t001:** Characteristics of patients enrolled in the study.

		ACL	Gonarthosis	Total
Gender	male	19 (11 adolescent)	9	28
	female	11 (5 adolescent)	9	20
	total	30	18	48
Age Median (IQR)		18 (5.50)	73 (11.0)	28.5 (51.75)
Effusion	hydrarthrosis	13	14	27
	mild hemarthrosis	2	1	3
	hemarthrosis	15	3	18
Timing	acute (0–2 days)	6 (20%)	0 (0%)	6 (12.5%)
	early subacute (3–15 days)	3 (10%)	0 (0%)	3 (6.25%)
	late subacute (16–90 days)	21 (70%)	0 (0%)	21 (43.75%)
	chronic (>90 days)	0 (0%)	18 (100%)	18 (37.5%)

**Table 2 biomedicines-10-02909-t002:** Assay ID used for RT-qPCR.

microRNA	Assay ID Number	Sequences
mirna 26a-5p	477995_mir	UUCAAGUAAUCCAGGAUAGGCU
mirna 27a-3p	478384_mir	UUCACAGUGGCUAAGUUCCGC
mirna let7a-5p	478575_mir	UGAGGUAGUAGGUUGUAUAGUU
mirna 140-5p	477909_mir	CAGUGGUUUUACCCUAUGGUAG
mirna 146-5p	478513_mir	UGAGAACUGAAUUCCAUAGGCU
mirna 155-5p	483064_mir	UUAAUGCUAAUCGUGAUAGGGGUU
mirna 16-5p	477860_mir	UAGCAGCACGUAAAUAUUGGCG
mirna 186-5p	477940_mir	CAAAGAAUUCUCCUUUUGGGCU
mirna 199a-3p	477961_mir	ACAGUAGUCUGCACAUUGGUUA
mirna 210-3p	477970_mir	CUGUGCGUGUGACAGCGGCUGA
mirna 205-5p	477967_mir	UCCUUCAUUCCACCGGAGUCUG
mirna 30b-5p	478007_mir	UGUAAACAUCCUACACUCAGCU

**Table 3 biomedicines-10-02909-t003:** Ct values for miRNAs measured by RT-qPCR.

microRNA	Value = 0 (*n*)	Value > 0 (*n*)
**mirna 26a-5p**	26 (54.2%)	22 (45.8%)
**mirna 27a-3p**	39 (81.25%)	9 (18.75%)
**mirna let7a-5p**	43 (89.6%)	5 (10.4%)
**mirna 140-5p**	42 (87.5%)	6 (12.5%)
**mirna 146-5p**	42 (87.5%)	6 (12.5%)
**mirna 155-5p**	34 (70.8%)	14 (29,2%)
**mirna 16-5p**	31 (64.6%)	17 (35.4%)
**mirna 186-5p**	17 (35.4%)	31 (64.6%)
**mirna 199a-3p**	45 (93.75%)	3 (6.25%)
**mirna 210-3p**	43 (89.6%)	5 (10.4%)
**mirna 205-5p**	43 (89.6%)	5 (10.4%)
**mirna 30b-5p**	23 (47.9%)	25 (52.1%)

**Table 4 biomedicines-10-02909-t004:** (**A**) Values of mean CT (with standard deviation) and median Ct. (**B**) Expression of microRNAs in ACL and knee OA.

**(A)**				
	**Mean *(DS)***	**Median**	**Mean *(DS)***	**Median**
**mirna 26a-5p**	3.04 *(4.18)*	0.41	4.28 *(8.09)*	0
**mirna 27a-3p**	1.91 *(5.28)*	0	5.60 *(11.3)*	0
**mirna let7a-5p**	1.27 *(4.66)*	0	1.07 *(3.38)*	0
**mirna 140-5p**	0.62 *(3.39)*	0	4.41 *(8.59)*	0
**mirna 146-5p**	1.69 *(4.96)*	0	0.68 *(2.43)*	0
**mirna 155-5p**	1.92 *(3.63)*	0	5.58 *(9.35)*	0
**mirna 16-5p**	2.55 *(3.90)*	0	3.26 *(5.42)*	0
**mirna 186-5p**	5.69 *(5.87)*	**6.00**	9.18 *(7.11)*	**9.47**
**mirna 199a-3p**	0.19 *(0.82)*	0	0.47 *(2.00)*	0
**mirna 210-3p**	1.20 *(6.57)*	0	3.87 *(9.26)*	0
**mirna 205-5p**	0.62 *(2.01)*	0	3.19 *(9.30)*	0
**(B)**				
	**ACL (*n* = 30)**		**Gonarthrosis (*n* = 18)**	
	**Value = 0**	**Value > 0**	**Value = 0**	**Value > 0**
**mirna 26a-5p**	15 (50%)	15 (50%)	11 (61.1%)	7 (38.9%)
**mirna 27a-3p**	26 (86.7%)	4 (13.3%)	13 (72.2%)	5 (27.8%)
**mirna let7a-5p**	27 (90%)	3 (10%)	16 (88.9%)	2 (11.1%)
**mirna 140-5p**	29 (96.7%)	1 (3.3%)	13 (72.2%)	5 (27.8%)
**mirna 146-5p**	26 (86.7%)	4 (13.3%)	16 (88.9%)	2 (11.1%)
**mirna 155-5p**	23 (76.7%)	7 (23.3%)	11 (61.1%)	7 (38.9%)
**mirna 16-5p**	19 (63.3%)	11 (36.7%)	12 (66.7%)	6 (33.3%)
**mirna 186-5p**	13 (43.3%)	17 (56.7%)	4 (22.2%)	14 (77.8%)
**mirna 199a-3p**	28 (93.3%)	2 (6.7%)	17 (94.5%)	1 (5.5%)
**mirna 210-3p**	29 (96.7%)	1 (3.3%)	14 (77.8%)	4 (22.2%)
**mirna 205-5p**	27 (90%)	3 (10%)	16 (88.9%)	2 (11.1%)
**mirna 30b-5p**	17 (56.7%)	13 (43.3%)	6 (33.3%)	12 (66.7%)

## Data Availability

Not applicable.

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
