# Peer review of "Characterization of microRNA Levels in Synovial Fluid from Knee Osteoarthritis and Anterior Cruciate Ligament Tears"

_biomedicines, 2022, doi:10.3390/biomedicines10112909_

Round 1

Reviewer 1 Report

In this study the authors quantified a series of microRNAs by Real-time quantitative polymerase chain reaction (RT-qPCR) in knee synovial fluid of patients with osteoarthrosis (OA) and rupture of the anterior cruciate ligament (ACL). They found some miRNAs differentially expressed.

The study can easily read and the results, although not very significant, are clear. The limitations of the study are clearly disclosed at the end of the manuscript.

I have one comment and one suggestion.

Although the graphs showing the quantification of microRNAs are easily to interpret, I did not understand the definition of Ct levels. Is it the Delta between 40 (number of PCR cycles) and the Ct (cycle threshold) of the real-time PCR? Please, explain better in the methods and possibly in the legend of the figures.

A log scale for the Ct values may be better to represent the results, because in some graphs, a single sample has a Ct value far higher than the others. The authors could try this option and then decide to accept or not the suggestion.

Author Response

The Ct (cycle threshold) is defined as the number of cycles required for the fluorescent signal to exceed the background level (i.e. the lower the Ct value, the more PCR product that is present). Ct levels are inversely proportional to the amount of target nucleic acid in the sample (ie the lower the Ct level the greater the amount of target nucleic acid in the sample).

We have followed the reviewer’s suggestion to represent the values in a logarithmic scale, but looking at the results we think that the log scale does not make them easier to read. For this reason, we prefer to keep the present graphs.

Reviewer 2 Report

Taking into account the epidemiology of cruciate ligament ruptures as well as that of arthrosis, the topic of the article is of interest. This is also due to a still small number of effective therapeutic and prophylactic options.

However, I have some concerns about the design of the study:

- why did you include the pediatric population (12-18 years old) - there are differences between the structure and quality of the articular and periarticular components at this age compared to adults

- the exclusion criteria mention previous traumas, in this context should we understand that the ligament tear is spontaneous? (rows 93-95)

- two small sample sizes

Also:

-I would prefer to read a longer introduction about miRNA as biological regulators.

-please include reference for Kellgren-Lawrence (row 101)

Author Response

- Pediatric patients were enrolled because ACL tears are continuously increasing in this population and clinical and preclinical data about these subjects are needed by Pediatric Orthopaedic Surgeons and Researchers. The biochemical environment in skeletally immature patients has been poorly studied and our results suggest that more pre-clinical studies are needed to better understand the healing potential of ACL tears in skeletally immature patients. To better clarify this point, as requested by the reviewer, we have modified methods, results and discussion sections.

There are differences between the structure and quality of the articular and periarticular components at this age compared to adults.

- Thank you for this comment. We have considered only traumatic ACL tears without a previous traumatic meniscal, chondral and ligamentous tear of the same limb as well as without a previous fracture of the interested limb. The text was revised per suggestions.

- In our database, we have new samples to examine that could confirm the data obtained.

- Approximately 2200 miRNA genes have been reported to exist in the mammalian genome, from which over 1000 belong to the human genome. Many major cellular functions such as development, differentiation, growth, and metabolism are known to be regulated by miRNAs.

- Thank you, reference was included as per suggestion.

Reviewer 3 Report

The main goal of the manuscript, that is to investigate a differential expression of miRNAs in the synovial fluid of patients with knee osteoarthritis or rupture ACL and to confront these two is very nice and ambitious. I find this story very positive and interesting and the attempts to identify molecular diagnostic molecules of OA is definitely needed. However, I cannot find the results presented here as convincing.

My main criticism concerns:

1.       the miRNAs chosen for the study

There are now more than 80 miRNAs described nicely in the literature to be somehow involved in OA development or/and progression. Why did the Authors chose to study the described 12 miRNAs? You should discuss that in Materials and methods section.

2.       The title

I do not agree with the title – the Authors did not identify miRNAs. They investigated the profiles of expression ( and NOT the “modifications of miRNAs expression profiles”, as stated in the abstract section). And came to the conclusion 10 of 12 tested miRNAs are present both in OA and ACL samples. And the conclusion about this result really surprises me. Didn’t you actually expected this result? Wouldn’t it mean that OA is already progressing after ACL rupture? I think this is it what is the most interesting here, but now the article states something different.

3.       Too strong statements

I do not see „the evidence that the differential expression of miRNAs in the synovial fluid of patients could be involved in the intraarticular regu-lation of the inflammatory pattern”, as you state in the Abstract section – please provide sufficient experiment or tone down your statement.

4.       The most importan results obtained here are not highlighted

1.       My first though was that the results in paragraph 3.1 are presented in a very strange way. However, I think this is the main clue of the paper. And this you may be willing to emphasis this result. A similar expression profiles of OA-related miRNAs in patients with ACL rupture might suggest a development of OA in ACL patients. And even more, the expression of the miRNA-30b - higher in OA and chronic ACL groups compared to acute ACL samples would add to this result.

However, to justify such results, it would be needed to include some control samples (optimally healthy ones, but it might be difficult to collect synovial fluid of a healthy person) to show that these miRNAs are not present in healthy samples as well.

Some other things, but also needed to be improved/changed:

2.       Line 46: One of the most important risk factors is knee trauma – Well, I wouldn;t say so. It’s the most important factor for post-traumatic OA development, but not OA in general. Please rephrase that.

3.       Line 101 – please specify which KL grades were taken into account. It is important for the group to be homogenous in that respect. One could speculate that different miRNAs might be expressed at different progression stage of the disease, right? And maybe this is the cause of the results that you observe.

4.       I do not quite understand the analysis of gender-relation to miRNA expression. Don’t you think that having just 9 females is not enough to generalize about gender influence? Did you caluclate a’priori what would be the minumum sample size to do so? I’d suggest to do so; if not - delete that paragraph.

5.       Why is miRNA-140 so interesting, as it appeared in just one sample out of 30?

6.       Table 2 brings nothing important to the paper, however presenting the sequences of the primers used for RT-qPCR would be of high importance.

In conclusion, I think that the manuscript has a nice potential, but the results are not highlighted. What would be my advise is to re-write the paper again, strenghten the things that are the most interesting, include the control samples, present materials nad methods better. I will be happy to re-review the manuscript after Author’s corrections.

Author Response

  1. It has been suggested that miRNAs might play a key role in OA initiation and progression. We chose to study these 12 miRNAs because it has been shown that they are involved in degenerative processes relevant to this study. In fact, bone metabolism is a multifactorial process that involves several cell populations such as bone marrow mesenchymal stem cells, osteoblasts, osteoclasts, as well as osteocytes. In these cells, there is the activation of key signaling pathways that can be modulated by modifications of the 12 miRNA levels selected in this research.
  2. Thank you. We accept your suggestion and we have modified the title as suggested. We think that this study may be considered as a starting point to better understand the real role of miRNAs after an ACL rupture, and we have tried to stress this point in the results and conclusion. It is very likely that dysregulation in miRNAs expression after ACL rupture could be part of the complex regulation pattern that leads to OA several years later. We have previously reported that inflammatory cytokines could also be involved in starting the processes that in the long term lead to OA
  3. Thank you, we have modified the final statement of the abstract as suggested.

  1. We have tried to emphasize as much as possible our results, as suggested. We also agree with the Reviewer that it would be very important to compare samples from OA and ACL with healthy control. However, performing an arthrocentesis in a healthy no swollen knee is unethical. In some studies, the synovial fluid used as control was obtained at autopsy, but there are some limitations also for the value of data obtained from these samples.
  2. Thank you for this comment. We rephrased the sentence to underline the multifactorial etiology of OA.
  3. Thank you, we have considered patients with a KL ≥ 3. The text was revised accordingly.
  4. In the study we enrolled forty-eight subjects (28 males and 20 females) that were divided into two groups based on their diagnosis. In particular, we had 9 males and 9 females in the gonarthrosis groups, and this sample size allowed optimal statistical power. For this reason, we could state (line 220) that our results demonstrate that gender did not affect the miRNA expression profile in the synovial fluid.
  5. miR-140 has been involved in the regulation of several important genes and it has been shown to be associated with the pathogenesis of a wide range of pathologies including osteoarthritis, osteoporosis, renal fibrosis, ischemic conditions, and most importantly neoplasia. miR-140 might have therapeutic potential in OA by regulating cartilage homeostasis and development.

Thank you for your comment, we have added the sequences of interest.

Round 2

Reviewer 3 Report

The authors have addressed most of my comments. Now the manuscript has increased the quality.